# How Many Secret Details Could a Systematic Multi-Analytical Study Reveal About the Mysterious Fresco *Trionfo della Morte*?

**Maria Francesca Alberghina** [1], **Salvatore Schiavone** [1], **Caterina Greco** [2],
**Maria Luisa Saladino** [3,*], **Francesco Armetta** [3], **Vincenzo Renda** [4] and **Eugenio Caponetti** [3,5]

1  S.T.Art-Test di S. Schiavone & C. S.A.S, Via Stovigliai 88, I-93015 Niscemi (CL), Italy
2  Museo Archeologico Regionale "Antonino Salinas", Via Bara all'Olivella 24, I-90133 Palermo, Italy
3  Dipartimento Scienze e Tecnologie Biologiche, Chimiche e Farmaceutiche - STEBICEF and INSTM UdR—Palermo, Università di Palermo, Viale delle Scienze pad.17, I-90128 Palermo, Italy
4  CNR – Istituto per I Processi Chimico-Fisici (IPCF), V.le F. Stagno D'Alcontres, 37, I-98158 Messina, Italy
5  LaborArtis C.R. Diagnostica s.r.l., Viale delle Scienze Ed.16, I-90128 Palermo, Italy
*  Correspondence: marialuisa.saladino@unipa.it

**Abstract:** The *"Trionfo della morte"* is a detached fresco painting dated at the half of the XV century. Its history is strictly connected with the history of Palermo and it is considered a symbol of the late Gothic period. Some small areas of the fresco were analyzed using a combination of non-invasive techniques and hand-held instrumentations (multispectral imaging analysis, X-ray fluorescence (XRF), and IR spectroscopy). The characterization of the nature of pigments used in its realization and restoration works was performed and some indications about its conservation state were obtained. More interestingly, some hidden details were revealed on the mysterious painting. They constitute additional evidence of the preciousness of the fresco.

**Keywords:** multispectral analysis; XRF; portable instruments; wall painting; *Trionfo della morte*

---

## 1. Introduction

The *"Trionfo della morte"* (Triumph of Death), as shown in Figure 1, is an imposing and unique fresco painting (600 × 642 cm) and is one of the greatest masterpieces of all time, with a shocking representative power [1]. An unknown painter realized the fresco indicatively between 1440 and 1450. In the mid-fifteenth century, Palermo was Hispanic and under the reign of Alfonso V d'Aragona known as the Magnanimous, when the spread of epidemics, famines, and the Black Plague changed the perception of death, until the formation of a new expressive sensitivity, which led to the birth of vast literature and an intense figurative development on his iconography. The *"Trionfo della morte"* thus became the most significant and emblematic artistic expression of the late Gothic period in Sicily. The fresco was realized in the courtyard of Palazzo Sclafani, built in 1330 at the behest of Count Matteo Sclafani, near the Palazzo dei Normanni, the Royal Palace. After Alfonso V d'Aragona's death and the long struggle for his succession, in 1400 the palace was confiscated and assigned to a noble Spanish family, who however, returned to Spain abandoning it to a progressive degradation. Thirty years later, the palace was chosen to be the first public hospital of Palermo and its courtyard has been decorated with sculptures, paintings, and frescoes, which in some way could accompany and alleviate, spiritually, the guests in their sufferings.

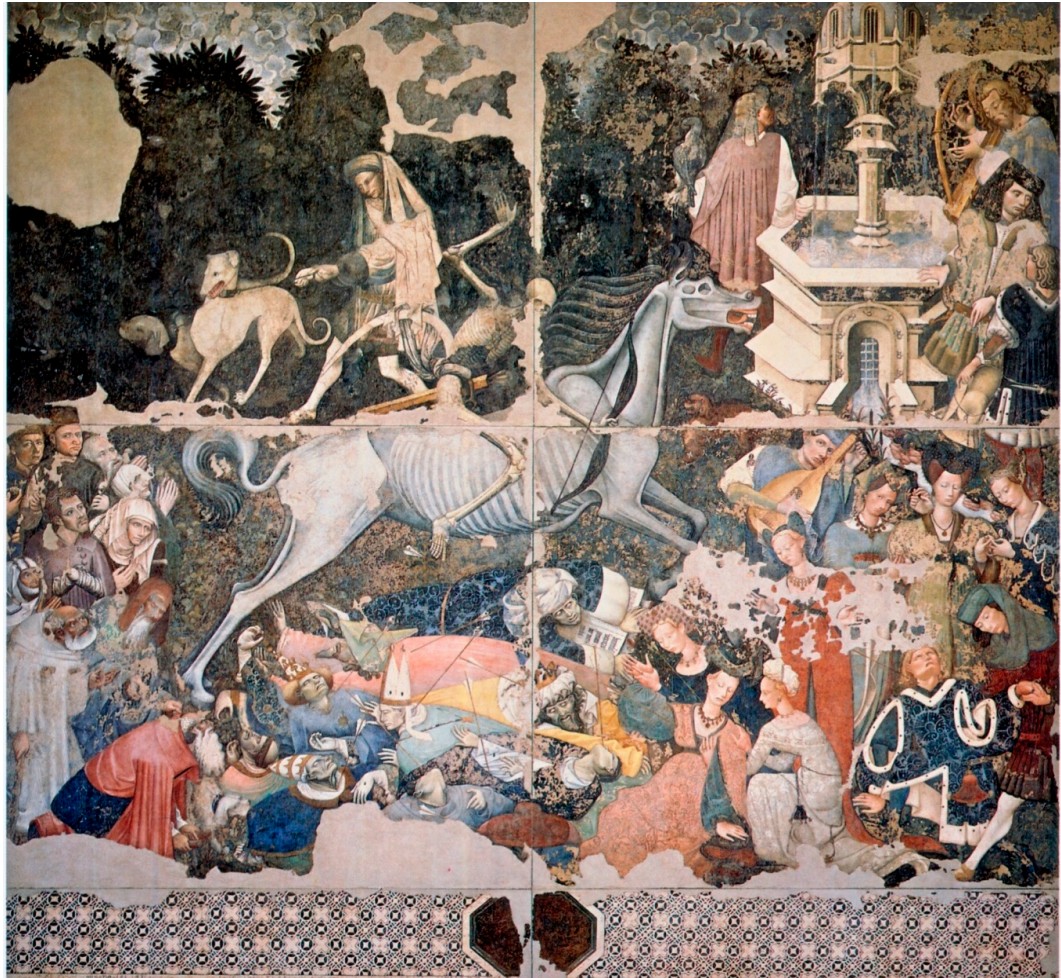

**Figure 1.** Photo of *Trionfo della Morte*, XV century, Gallery of Art for the Sicilian region Abatellis Palace, Palermo (Italy).

The large fresco remained in the courtyard of the hospital for five centuries, and it remained in almost perfect condition until the outbreak of the Second World War.

The bombing of Palermo in 1943 caused the collapse of the vaulted roof and other serious damage to Palazzo Sclafani; therefore, in order to preserve the masterpiece it was decided to detach it from the wall and move it to a safer place. Considering the size, the fresco was divided into four parts in order to detach it from the wall more easily. It was saved at Sala delle Lapidi of Palazzo Pretorio until 1954 when it was moved to the Regional Gallery of Sicily in Palazzo Abatellis (Palermo, Italy) where even now it is exposed in the new museum layout designed by the Venetian architect Carlo Scarpa.

The fresco was restored by the *Istituto Centrale del Restauro (ICR)*, under the care of Cesare Brandi. A further restoration in the 1980s was carried out by restorers Carlo Giantomassi and Donatella Zari under the supervision of Giovanni Urbani of the ICR. The restoration provided for the cleaning of the paintings and the application of new supports.

Today, the *"Trionfo della morte"* is considered as an "iconographic encyclopedia", a medieval allegory containing the essence of everything: life and death, beauty, old age, ease and poverty, sadness, bitterness, amazement, disdain, care, compassion, love, and hope.

Considering the uniqueness, the peculiarity, and the vicissitudes of this painting, it was deeply studied by a historical point of view, but, to our knowledge, no scientific investigation about the materials constituting the painting was ever performed. Therefore, we thought that a scientific study could be crucial to reveal invisible details, hidden by the artist or by time [2,3] to give additional information about its conservation state.

In this paper, results obtained by the application of multi-analytical non-invasive investigations carried out on a few areas of *"Trionfo della morte"* are reported.

The diagnostic campaign was carried out by following a well-consolidated approach applied to paintings starting from the documentation step by multispectral analysis and followed by the spectroscopic investigation [4–9]. Multispectral imaging (in UV, visible and infrared range, and false color infrared, FCIR), X-ray fluorescence (XRF), and Fourier transform infrared (FT-IR) spectroscopy were applied using hand-held instruments in order to investigate the pigments belonging to different figures and to identify the materials and methods used in the original realization and in the restoration. Some photos, taken during the campaign, are reported in Figure 2. The results here reported constitute a preliminary phase of the project study proposed to characterize the materials and the painting technique of the whole fresco surface, among the most important in the panorama of mural painting of the XV century in Sicily.

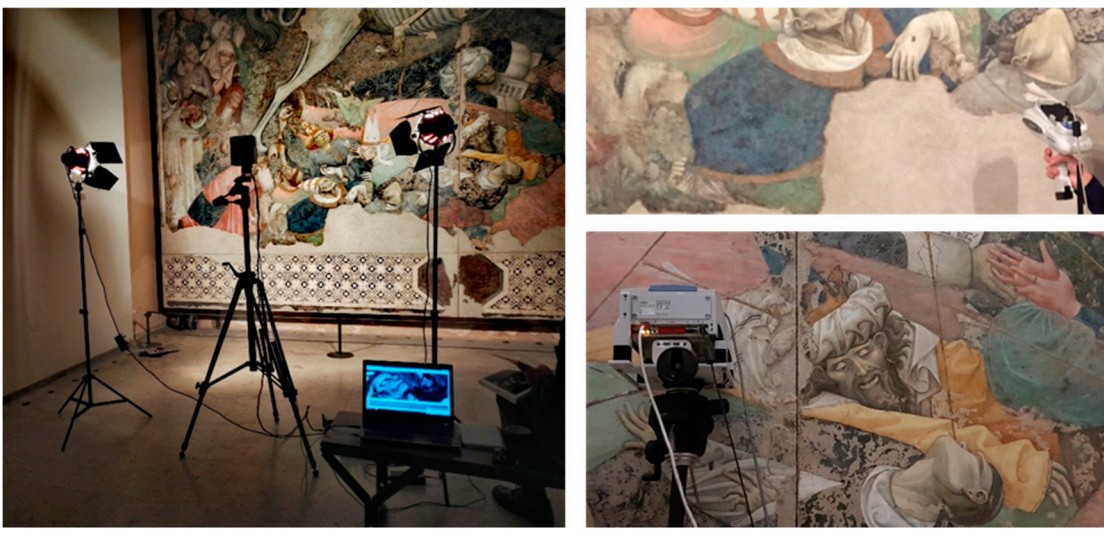

**Figure 2.** Photos of the in situ diagnostic campaign. (**left**) multispectral imaging acquisition; (**right-up**) XRF spectra acquisition; (**right-down**) IR spectra acquisition.

## 2. Instrumentations and Methods

The multispectral imaging acquisition was carried out using a digital CCD cooled scientific camera (CHROMA C4-DSP series, model C250ME by DTA srl) with a KAF8300ME sensor, equipped with an internal wheel containing interference filters (350, 400, 450, 540, 600, 750, 1000 nm bands filter, 50 nm bandwidth). The digital scientific camera uses a Cooled CCD, 6 Megapixel (3326 × 2504 points) effective (pixels with side of 5.4 μm). The acquired visible and infrared images were also processed to obtain infrared false color information (FCIR) for a preliminary mapping of different pictorial layers and for discriminating original or restoration surfaces.

XRF spectra were acquired with a Tracer III SD Bruker AXS instrument having a rhodium anode (Voltage 40 kV and current 11 μA) as source and a Silicon Drift XFlash®as a detector. Each spectrum was acquired for 30 s in an area of 3 mm$^2$ by placing the instrument in contact with the surface of the painting. The collected spectra were analyzed using the software ARTAX 7. In all acquired spectra, the signals of argon (Ar), rhodium (Rh), and palladium (Pd) elements are present due to the environment and the setup of the used device.

Reflectance FT-IR spectra were acquired using the portable Bruker ALPHA FT-IR Spectrometer equipped with an External Reflection QuickSnapTM module. Spectra were acquired between 3500 and 360 cm$^{-1}$ with a resolution of 4 cm$^{-1}$, and averaging 60 scans for each measurement, in an area of 5 mm$^2$, by placing the instrument in a contactless manner with the painting's surface with a working distance of about 15 mm. All reflectance spectra were processed using OPUS 7.5 software; every

spectrum was converted to pseudo-absorbance (log R) units in order to compare the spectrum with a database of reflectance reference spectra [10].

## 3. Results and Discussion

The application of the multispectral non-invasive investigation to the two pictorial areas of the "*Trionfo della morte*" fresco allowed the discovery of some iconographic details no longer visible to the naked eye and to map the precious "*a secco*" details. Some repainting areas are well evident because of the use of "*rigatino*" pictorial integration techniques. Nevertheless, thanks to different spectral behaviors, the original layers are immediately distinguishable from pictorial integrations not distinguishable to the naked eye, not even to a close view.

Some of the multispectral images obtained by the multispectral investigation (visible, FCIR, and IR reflectography) are reported in Figures 3–8.

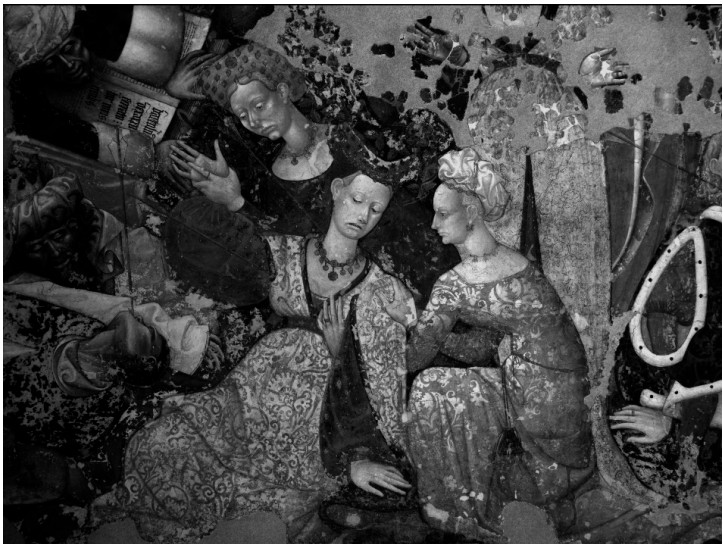

**Figure 3.** Infrared reflectography image acquired with a 1000 nm filter on a particular of the female figure group.

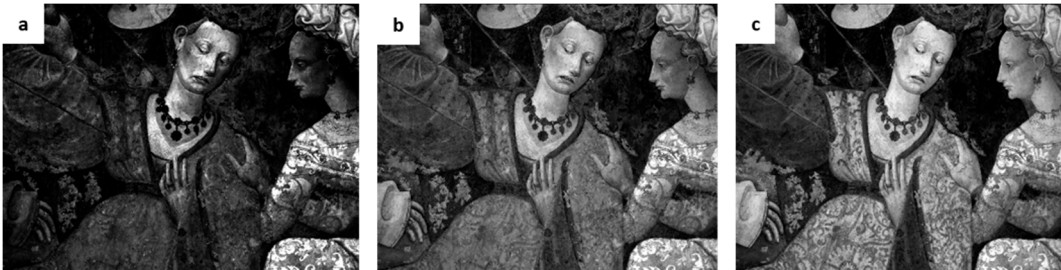

**Figure 4.** Visible range images acquired with (**a**) 450 nm, (**b**) 540 nm, (**c**) 600 nm filters on the detail of the female figure group.

The first analyzed area, corresponding to the group of women in the foreground at the bottom of the fresco, as shown in Figures 3–5, confirmed the importance of a systematic imaging analysis on this work of art. It is in fact clearly visible the preciousness and complexity of the iconographic painted subjects that become again readable thanks to the pictorial surface observation in the single filtered spectral bands.

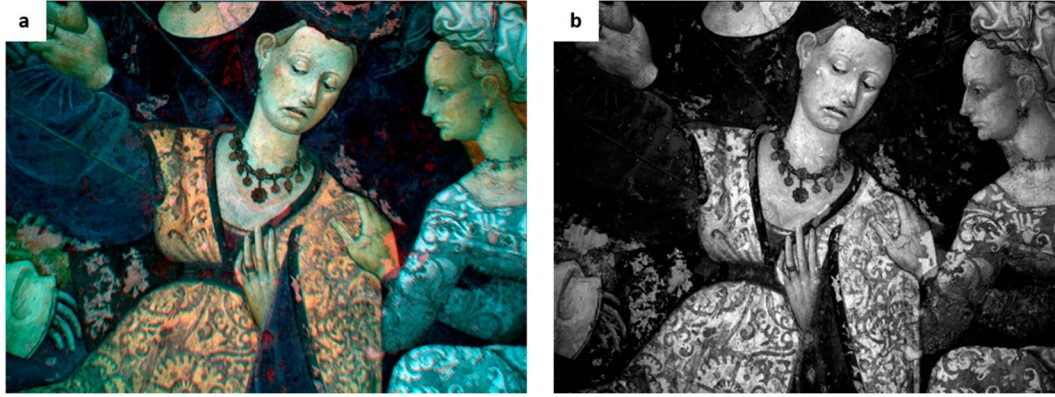

**Figure 5.** Comparison between (**a**) the false color infrared (FCIR) image and (**b**) the IR reflectography (1000 nm) on the detail of the female figure group. The characteristic spectral color provided by FCIR acquisition suggests the typology of the used pigments for original or restoration areas. For example, the green area of the dress could be typical of copper-based green pigment (malachite) and the red area could be attributable to chromium green constituting the localized integration in pictorial layers.

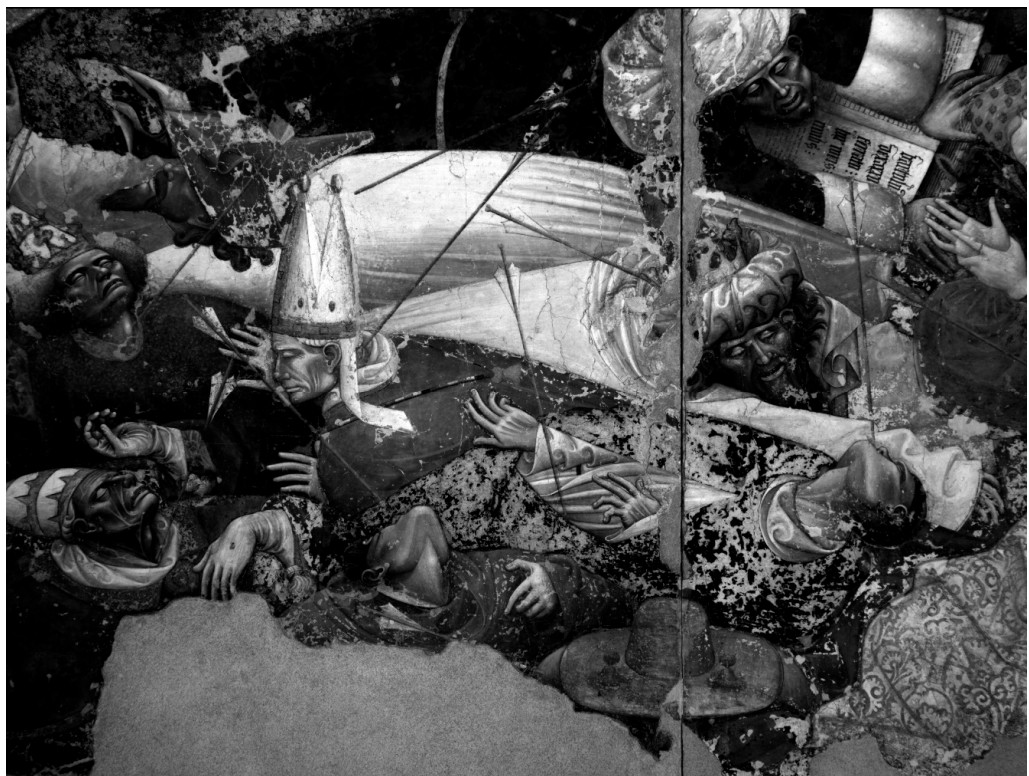

**Figure 6.** Infrared reflectography image acquired with a 1000 nm filter on a particular of the lower central part of frescos.

In some cases, the FCIR images provided preliminary information about the pigments on the basis of IR spectral behavior, and, thanks to the comparison with FCIR images, the XRF results were extended within the whole painted area characterized by the same spectral color. For example, as observable in Figure 5, the yellow tone of the red dress of the woman suggests the presence of cinnabar used in the mixture due to the typical spectral response of this red pigment in FCIR imaging analysis [11–13].

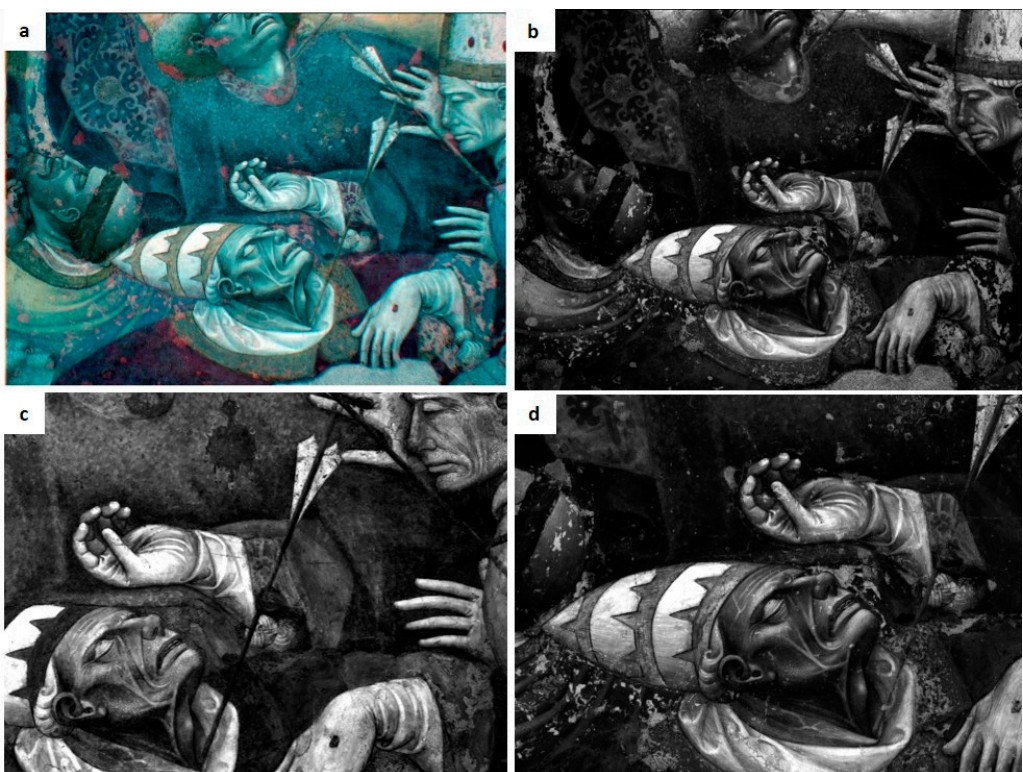

**Figure 7.** Comparison between (**a**) the false color infrared (FCIR) image and (**b**) the IR reflectography (1000 nm) on the detail of the bishop. (**c**) Details of the beard "*a secco*" of the bishops; (**d**) the IR reflectography zoomed images highlight the path of "*giornata*" frescos along the contour of the bishop's figure (face and hand).

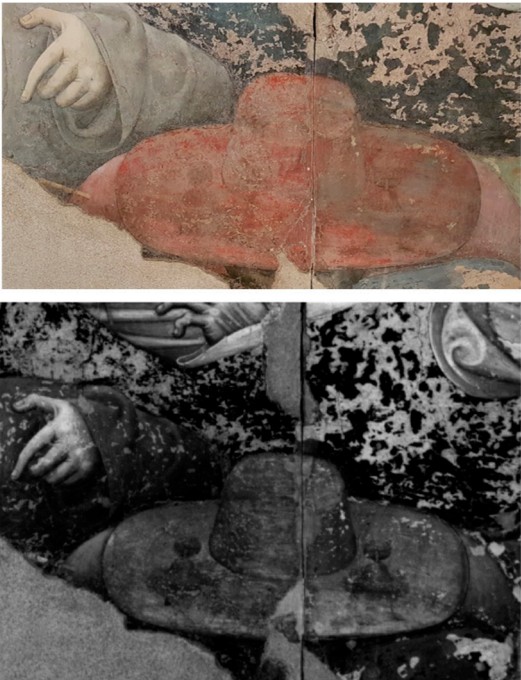

**Figure 8.** Comparison between (**up**) the visible image (**down**) the IR reflectography (1000 nm) on the detail of the red hat. The hidden decorations of the red hat are readable in the IR image.

The details of the two women's collars and the headgear of the central figure are more noticeable in the filter visible range, which became transparent in infrared, as shown in Figures 3–5, and consequently

identifiable as "*a secco*" details. On the contrary, the fresco surface is not transparent to the IR spectral range due to the high IR absorption of calcium carbonate matrix [9]. Under a UV lamp a yellowish fluorescence in correspondence to the "*a secco*" decorations was observed. This evidence could be associated to a resinous oleo material used as binder for pictorial "*a secco*" layers [14,15].

Moreover, IR reflectography of the central female figure highlights the breast contour, hidden by a white-collar decoration in visible range, as shown in Figures 3 and 5b.

Also, in the second investigated area, as shown in Figures 6–8, the multispectral imaging provided a clearer legibility of iconographic detail and allows us to provide preliminary information about the executive techniques of pictorial layers.

For example, the details of the "*a secco*" arrows, almost transparent to the IR reflectography, allow us to see the underlying drapery, as shown in Figure 6. The beard "*a secco*" details of the bishops become much more appreciable with a 450 nm filter, as shown in Figure 7c.

Moreover, the contours of the "*giornata*" frescoes are more evident along the profile of the bishop's face in the IR range, as shown in the detail in Figure 7d.

Finally, the decorations of the red hat, no longer appreciable to the naked eye, appear readable in the IR image thanks to the IR transparency of the red pigment, as shown in Figure 8.

The XRF and IR spectra were acquired in situ to investigate the nature of pigments, both original and used in restoration in order to provide preliminary information on the unknown artist's palette and, then, on the typical materials and executive techniques of the XV century Sicilian artistic production still not systematically studied from an archaeometric point of view. The points for spectra acquisition were chosen on the basis of the multispectral investigation, distinguishing the different materials thanks to the different spectral behavior. The maps of the points analyzed by XRF are reported in Figure 9. Measuring points' numbering, color hatch together with a short description of the analyzed points and the identified elements are reported in Table 1. Some of the XRF spectra are reported in Figures 10–12.

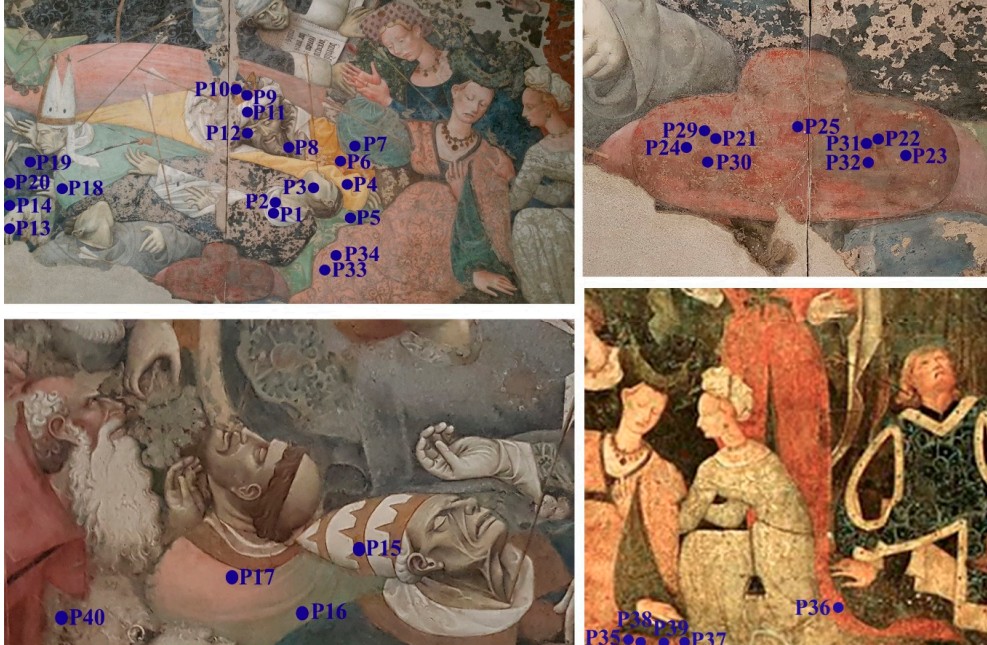

**Figure 9.** Location of the X-ray Fluorescence (XRF) measuring points in some area of the painting.

**Table 1.** Measuring points' numbering, color hatch together with a short description of the analyzed point and the identified chemical elements. The abundant elements are reported in bold.

| Point | Color | Identified Chemical Elements |
|---|---|---|
| 1 | white vest | S, **Ca**, Fe, Sr, Pb |
| 2 | grey vest | K, **Ca**, Ti, Fe, Sr, Pb |
| 3 | incarnate friar | **Ca**, Fe, Sr |
| 4 | yellow sleeve | **Ca**, **Fe**, Sr, Sn, **Pb**, Au |
| 5 | green next to yellow | K, **Ca**, Cr, Mn, **Fe**, Sr, **Pb**, Sn, Au |
| 6 | orange | Si, K, **Ca**, Ti, **Fe**, Sr, Pb |
| 7 | dark green | Si, P, K, Ca, Cr, Mn, **Fe**, **Cu**, Sr, Pb |
| 8 | brown beard | Si, K, **Ca**, Mn, **Fe**, Sr, **Pb** |
| 9 | blue hat | K, Ca, Cr, Mn, **Fe**, **Cu**, Sr, Pb, Ba |
| 10 | yellow hat | Si, S, K, **Ca**, Ti, Cr, Fe, Sr, Au |
| 11 | white turban | S, **Ca**, Fe, Sr |
| 12 | black turban | Si, S, K, **Ca**, **Fe**, Sr |
| 13 | hand of bishop | Si, S, **Ca**, Fe, Sr |
| 14 | blue mantle | Si, S, K, **Ca**, Cr, Mn, **Fe**, **Cu**, Sr, Ba, Pb |
| 15 | yellow hat | Si, P, S, K, **Ca**, Ti, Cr, **Fe**, Sr, Pb |
| 16 | green water | Si, P, K, **Ca**, Cr, Mn, Fe, Sr |
| 17 | pink mantle | Si, P, S, **Ca**, Fe, Sr |
| 18 | green mantle | Si, S, K, **Ca**, Cr, **Fe**, Sr |
| 20 | grey light | Si, S, **Ca**, Fe, Sr |
| 22 | red right pendant | Si, S, K, **Ca**, Cr, Mn, **Fe**, Sr, Ba, **Hg**, **Pb** |
| 24 | red hat | Si, S, K, **Ca**, **Fe**, Sr, **Hg**, Pb |
| 25 | red hat | Si, S, K, **Ca**, **Fe**, Sr, **Hg**, Pb |
| 26 | red low pendant | Si, S, K, **Ca**, Cr, Mn, **Fe**, Sr, **Hg**, Pb |
| 27 | red right low pendant | Si, S, K, **Ca**, Ti, Cr, Mn, **Fe**, Sr, **Hg**, Pb, Ba |
| 28 | integration red hat | Si, S, **Ca**, Cr, Mn, Fe, Sr, Pb |
| 33 | pink mantle | Si, P, S, K, **Ca**, Cr, Fe, Sr, Pb |
| 34 | grey decoration | Si, P, S, K, **Ca**, Fe, Sr, Pb |
| 35 | red mantle | S, K, **Ca**, Cr, Fe, Sr, Hg, Pb |
| 36 | red stole | Si, P, S, K, **Ca**, Mn, **Fe**, Sr, Pb |
| 37 | red mantle | S, K, **Ca**, **Fe**, Sr, Hg, Pb |
| 40 | red left figure | Si, S, K, **Ca**, **Cr**, Mn, Fe, Sr |

In analyzing the spectra, it should be taken into account that, due to the penetration depth of X-rays, the signals are related both to the pigments and the wall support. All XRF spectra show the signals of calcium (Ca), ascribed to the presence of calcite ($CaCO_3$) and/or gypsum ($CaSO_4 \cdot 2H_2O$). The presence of sulfur (S) and strontium (Sr) in high quantities suggests the presence of gypsum as well as celestite ($SrSO_4$), a mineral usually found associated with gypsum. More precise information could be obtained by using a portable X-ray diffractometer for the investigation of minerals. The presence of gypsum can be an alteration product of calcium carbonate or a component deliberately used by the artist [16,17]. The presence of strontium was connected to the last reason. On the other hand, the celestite in Sicily is often present in volcanic deposits from Mount Etna, but also can be commonly present in gypsum derived from seawater.

The widespread presence of lead (Pb) in some analyzed areas, in smaller quantities than calcium, is indicative of the presence of **white lead** (basic lead carbonate $(PbCO_3)_2 \cdot Pb(OH)_2$) used in the realization of the details "*a secco*". It is well known that the fresco techniques are not compatible with the presence of white lead [9] because of its reactivity in the basic environment. Its presence is thus evidence of the "*a secco*" details (confirmed in some cases by the IR transparency obtained in the corresponding IR reflectography) probably applied by using an oil as binder, as verified under UV lamp examination.

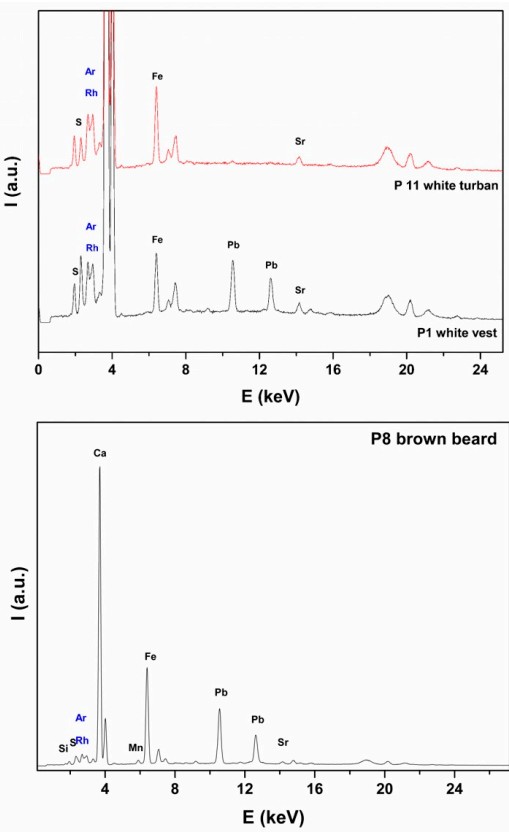

**Figure 10.** XRF spectra of some white and brown analyzed points/colors.

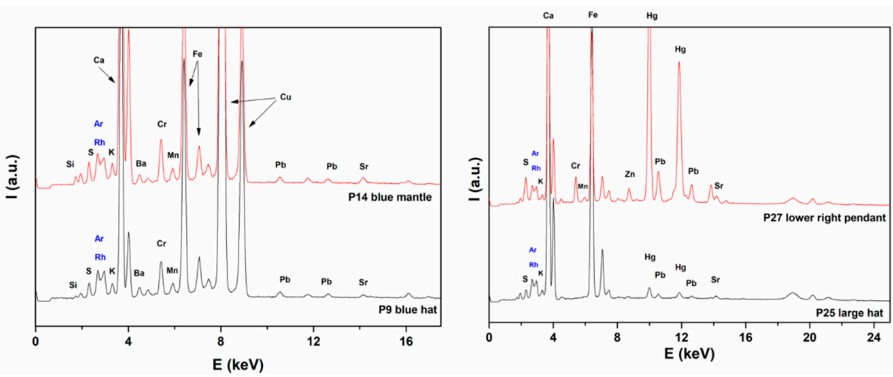

**Figure 11.** XRF spectra of some analyzed blue and red points/colors.

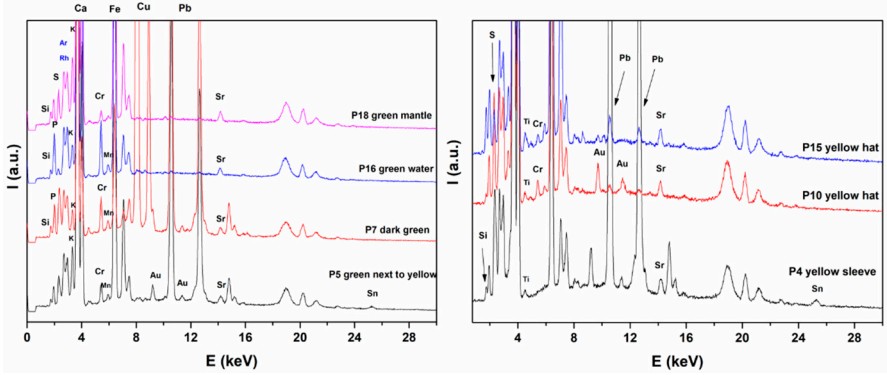

**Figure 12.** XRF spectra of some analyzed green and yellow points/colors.

On the other hand, the P1 and P11 spectra of white mainly show the signals of Ca together with small signals of lead (Pb), indicating that the white is **calcium carbonate.** In addition, the IR spectrum of the white shows a sequence of peaks matching with calcite and gypsum. The pseudo-absorbance IR spectrum obtained by taking the negative logarithm of the reflectance data acquired (-log R) is reported in Figure 13.

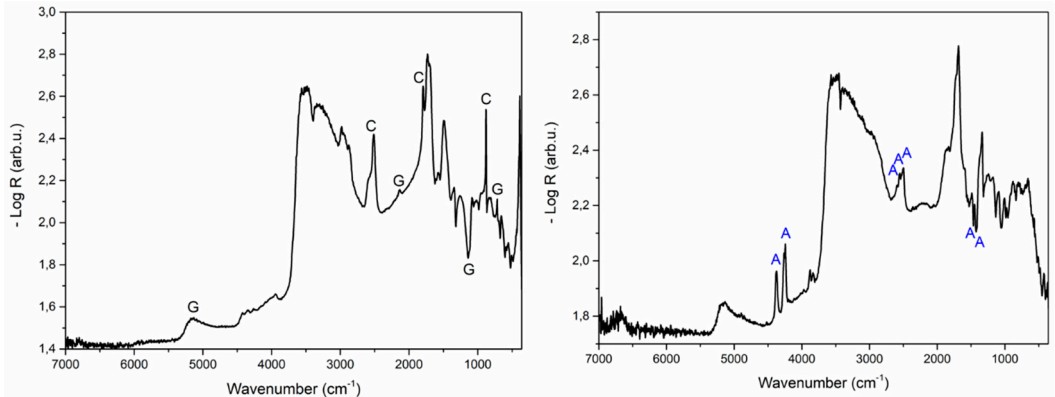

**Figure 13.** IR spectra of (**left**) white (C = calcite, G = gypsum) and of (**right**) blue (A = azurite).

The absorbance and derivative peaks were identified, by comparison with a series of pseudo-absorbance spectra obtained on standards, as gypsum and calcite and are highlighted in the figure (calcite C, gypsum G). Regarding calcite, in reflection mode, the lowest absorption due to symmetric carbonate stretching around 1000 cm$^{-1}$ shows a typical derivative appearance; at 1800 cm$^{-1}$ a very sharp and characteristic absorption due to a $v_1 + v_4$ combination is present. Calcite has also a signature and a very wide band at 2512 cm$^{-1}$, with a shoulder at 2594 cm$^{-1}$ due to $v_1 + v_3$ mode coupling. Gypsum presents signals which are overall less traceable in respect to the calcite signals. The lowest signal, under 1000 cm$^{-1}$, can be attributed to bending mode vibrations. Due to the higher absorption index, $v_3$ antisymmetric stretching of sulphate is affected by *reststrahlen*, having its minima around 1160 cm$^{-1}$. The weak peak emerging near 2200 cm$^{-1}$ is due to a combination band of bending and vibration modes of H$_2$O ($v_2 + v_L$). The last band in the NIR zone, around 5140 cm$^{-1}$, can be attributed to the mixing of vibrational modes connected to OH bonds of hydration water ($v_1/v_3$ OH + $v_2$ OH). The presence of gypsum could be due to a sulphating process, common degradation in this typology of painting. The presence of silicon (Si) and potassium (K), characteristic of silicates, can be due to some impurities of the chromophores or aggregate in the support layers. These signals can be related to both the pigments and the wall support. Traces of titanium (Ti) were recognized in P2 and P6 indicating the possible use of titanium dioxide as white pigment in some areas of pictorial repainting. The high signals of iron (Fe) present in most of the spectra indicate the extensive use of iron oxide-based pigments, **ochres**, or earths which can be in several colors and are not expensive. Some considerations about the nature of the pigments are presented and discussed for each color.

**Black and grey:** Due to the absence of signals in the XRF, black (P12) pigment can be probably constituted by carbon black which is not identifiable by the XRF technique. The grey (P20) was obtained by mixing the black pigment with an appropriate amount of calcium carbonate to obtain the desired tonality. The vest (points P2 and P34) was realized by using white lead.

**Yellows:** The signals of iron (Fe) are present in both XRF spectra of the P10 and P15 indicating the use of yellow **ochres**. The low counts of the chromium signal could be attributable to the presence of the modern pigment chrome yellow (PbCrO$_4$) used for little retouching during the previous restorations. The signals of lead (Pb) and tin (Sn) are present in the P4 spectrum of the sleeve indicating the presence of the lead-tin yellow (Pb$_2$SnO$_4$ or PbSnO$_3$). This yellow pictorial area is characterized by a totally different IR spectral behavior (light grey tone) with respect the previous analyzed area (P10 and P15) which returned a greater absorption in IR (dark grey tones), as shown in Figures 6 and 7. This evidence

confirms the advantages of the integrated use of multispectral imaging and spectroscopic techniques, extending the result of chemical identification on a single point to extensive areas characterized by the same spectral behaviors. Furthermore, the obtained results underline the variety of the artist's palette (more materials, such as yellow ochres and lead-tin yellow for the yellow layers) to obtain different tonal effects for the same color. Finally, the small peak of **gold** (Au) in P4 and P10 spectra, can be related to some decoration in the drapery, confirming the preciousness of the fresco and providing important evidence of the original feature.

**Green**: The signals of chromium (Cr) in the XRF spectra of the P5, P7, P16, and P18 indicate the use of chromium (III) oxide, used for pictorial integration as confirmed by multispectral findings. However, the signal of phosphorus (P) is present in the spectra P7, P16, and P18. Generally, the presence of this element in fresco surfaces is attributable to the trace of the organic chemical compounds applied over time as protective identification. A further phase of the research project will involve a GS-MS analysis to exactly identify the original or restoration materials overlapped on these precious surfaces. The high Cu signal intensity revealed that the original dark green layers (P7), according to the spectral behavior in FCIR as shown in Figure 5, is identifiable as malachite [$Cu_2(CO_3)(OH)_2$] or verdigris [copper acetate] [11–13], not unambiguously discriminable through the non-destructive techniques used to date for the study of the here investigated fresco.

Brown: The presence of iron (Fe) and manganese (Mn) in the P8 spectrum indicative of umber (oxide $Fe_2O_3 \cdot H_2O$ hydrated iron with manganese dioxide $MnO_2$) and clay silicates like kaolin ($Al_2O_3 \cdot 2SiO_2 \cdot 2H_2O$) or burnt umber (anhydrous iron oxide $Fe_2O_3$ and manganese dioxide $MnO_2$).

**Flesh tone**: The intensive XRF signals of calcium (Ca), iron (Fe), and strontium (Sr) in P3, P13, P17, and P33 spectra indicate a combination of calcium carbonate and red ochres to reach the desiderated tonality.

**Blue:** The XRF spectra of the P9 and P14 show the signals of copper (Cu). In addition, the pseudo-reflectance spectrum reported in Figure 13 shows a sequence of peaks attributable to the azurite ($Cu_3(CO_3)_2(OH)_2$, letter A in the spectrum). In reflection mode, the strong absorption due to the carbonate antisymmetric stretching ($\nu_3$) in the region 1450–1420 cm$^{-1}$ is inverted by *reststrahlen* effects. In particular, azurite, in a different way in respect to other types of carbonates like cerussite and malachite, presents a fine structure of two inverted peaks for this vibrational mode. The second fine structure occurs around 2800–2300 cm$^{-1}$, whose possible attribution is due to coupling of $\nu_1 + \nu_3$ or $2\nu_2 + \nu_4$ modes. Moreover, the three distinct contributions between 2500 and 2600 cm$^{-1}$ and the third fine structure at a higher wavenumber, in the NIR range between 4200 and 4400 cm$^{-1}$, referable to both combination $\nu + \delta$ (OH) and overtone $3\nu_3$, are characteristic of azurite [18]. The high signals of iron (Fe) together with the small signals of manganese (Mn), silicon (Si), and sulfur (S) in the XRF spectrum indicate the presence of primer realized "on bolo" used for the spreading of the azurite. The low signals of chromium (Cr) and barium (Ba), in the spectrum can be attributed to some repainting made on the surface layer to give tonal homogeneity to the original blue layer. The presence of a veiling retouching layer was evidenced by the FCIR which returned a reddish spectral response superimposed on the typical azurite response in false color infrared, as shown in Figure 7.

Red: The investigation about red colors has been performed mainly in the red hat at the bottom figure where some blackening is present, and the reflectometry revealed some hidden decorations, as shown in Figure 8. The signals of iron (Fe) are present indicating the use of red ochre. The signals of lead (Pb) are also present in the P22–P40 spectra indicating the presence of lead oxide, generally known as minium. However, the signals of mercury (Hg) are present in the P24, P25, P26, P27, and P35 spectra. Results are in agreement with the IR reflectography by which the transparency of cinnabar at 1000 nm highlights the lost pendants (both at left and right). The blackening of red can be explained as the well know degradation of cinnabar [19,20]. The P27 spectrum revealed the presence of zinc (Zn) and chromium (Cr) attributable to zinc white (ZnO) and chrome red ($PbCrO4 \cdot PbO$), respectively, added to surface during the pictorial retouches of this area.

## 4. Conclusions

In this paper, the results of an in situ diagnostic campaign carried out on a few small areas of the famous detached painting "*Trionfo della morte*" saved at the Gallery of Art for the Sicilian region Abatellis Palace in Palermo (Italy) are reported. Thanks to the non-invasive investigations, such as multispectral analysis, XRF, and IR spectroscopies carried out step-by-step with a critical and reasoned approach, it was possible to identify the nature of pigments to map the precious "*a secco*" details, to obtain some indications about:

1) The conservation state, in particular linked to the thinning ("*a secco*" layers) or chromatic alteration (blackening of cinnabar red layers) of some pictorial details which now prevents the appreciating of some iconographies;

2) The areas of previous restorations, in particular the integrations performed mimetically and therefore not directly distinguishable to the naked eye;

3) Some interesting hidden details, appreciable only through single spectral bands in the visible (450 nm, for on surface "*a secco*" layers) or in the infrared (1000 nm, for underlying layers) spectral range.

The identified original pigments were red, brown, red and yellow ochres, umber, cinnabar, azurite (overlapped on iron oxide-based pigment), green earth, minium, lead-tin yellow, lead white, and gold. In particular, a systematic XRF mapping of the trace of gilding could give back the preciousness of the original appearance of the decorations.

Some of the several hidden details discovered in the two figures of the women and along the profile of the bishop's face as well as in the red hat of a cardinal are very interesting and constitute additional evidence of the preciousness of the fresco, rich in details despite the fact that for a long time it was placed outside.

Nevertheless, the disappearing of these details is due to time and to the nature of materials used together during the various restorations. Moreover, the possibility of retracing the conservative history through the localization of the different pictorial integration methods permits the highlighting of the complex history of the past restorations. It has to be underlined that only a few small areas were investigated. Even though the total surface of the investigated areas represents a small part of the whole fresco, this study gives a significant contribution to the deepening knowledge of this masterpiece, and it can be useful for an informed conservation and restoration.

The finds confirmed the use of a typical pigment palette available in the XV century and also of the technical devices generally used by fresco painters from other areas of Italy. A complete diagnostic study will allow the widening of knowledge and to establish more precise comparisons based on objective data and used raw materials with contemporary schools not only in the Sicilian regional territory but also in the central Italian ones. In this way, a comparative archaeometric study could propose solutions for identifying the local or foreign artist, currently unknown, that in the XV century realized in Palermo, that this work famous all over the world.

Moreover, this study can represent a very effective example to demonstrate how useful and productive a multi-analytical scientific approach by using complementary spectroscopic techniques can be. In fact, in some cases, the nature of each pigment can be defined by only integrating the results coming from each technique. In addition, it aims to the dissemination and valorization of cultural heritage and, in our opinion, it constitutes a base and an incentive to perform a complete scientific study on the whole fresco.

*NOTE.* The *"Trionfo della morte"* has been focused on by a historical, artistic, as well as a scientific point of view, during the exhibition *Arte è Scienza 2017* promoted by *AIAr* [21], which involved simultaneously 13 sites in Italy. This work of art has proven to represent a very effective example for demonstrating how a multidisciplinary approach aimed at the dissemination and valorization of cultural heritage can be useful and productive. The visitor participated in a (always) new discovery that provided the interpretation of the analytical data, accompanying them in an experience that is very different from the traditional museum visit.

**Author Contributions:** Methodology, M.F.A. and M.L.S.; data analysis, S.S., V.R. and F.A.; data curation, S.S. and F.A.; writing—review and editing, M.L.S. and M.F.A.; supervision, E.C., C.G.

**Funding:** This work is part of the project "Development and Application of Innovative Materials and processes for the diagnosis and restoration of Cultural Heritage—DELIAS"—PON03PE 00214 2 (Programma Operativo Nazionale Ricerca e Competitività 2007–2013).

**Acknowledgments:** Thanks to the Director of Galleria Interdisciplinare Regionale della Sicilia di Palazzo Abatellis, Palermo (Italy) for having allowed to carry out the investigations.

**Conflicts of Interest:** The authors declare no conflict of interest.

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
