# Peer review of "How Many Secret Details Could a Systematic Multi-Analytical Study Reveal About the Mysterious Fresco Trionfo della Morte?"

_heritage, doi:10.3390/heritage2030145_

Round 1

Reviewer 1 Report

Dear authors, 

I have proofly read your manuscript. 

Whereas I appreciate the beautiness of the analysed fresco and the level of details (in terms of spatial resolution) of the collected multispectral images, I found that your manuscript lacks of novelty in terms of the employed scientific methods. 

In fact, it is based on the combination of hyperspectral imaging (spanning the visible spectral range), XRF and FTIR spectroscopies - achieved with the aid of commercial devices - but in recent years this protocon has been employed in different case studies. 

Possibly, multivariate analysis methods (as PCA) could be of help for strengthening the information detected with your multispectral camera and could allow one to better visulize hidden details in the fresco subareas. The generation of similarity maps could be helpful for extending the material identification provided by XRF and FTIR spectroscopies on the whole analysed portion of the fresco surface.

Furthermore, the discussion of the results of scientific analysis is limited in the manusript, whereas authors mainly report the results in terms of collected spectra and identified elements. It could be of interest to correlate the results provided by the different techniques. Also, it could be of interest to discuss the identified pigments in the context of the typical painting materials employed in that historical period.

I must admit that in some points the manuscript looks more like a report of scientific analyses applied on a specific artwork than a manuscript reporting some novel findings on a case study or some novel finding on novel analytical methods.

I'm sorry for my criticism, but I hope it can be helpful for improving your manuscript.  

Author Response

Thank you for your suggestions and help to improve our manuscript.

The goal of our work has been to demonstrate that it was possible obtain several information on a so big and varied fresco painting.

Thanks this encouraging preliminary data, we would like to extend our study to the entire painting in collaboration to the regional government and the Abatellis Museum. In this case, collecting more spectroscopic data and multispectral imaging at higher sampling frequency (by using more filters with less bandwidth, from UV to IR range) we will apply addition data treatments like the multivariate analysis to improve the understanding of the complex distribution of pictorial materials overlapped over time.

In order to improve the quality of the manuscript, we added some details about the data acquisition in the experimental part and some comments about the identified materials (original and not).

A brief additional description of the pigments and some general considerations have been added also in the conclusion.

All modifications in the manuscript have been reported in red.

Best Regards,

Maria Luisa Saladino

Reviewer 2 Report

This study is interesting. Results and discussion are well-described. I would strongly suggest to add a table, maybe in the Conclusions paragraph, with the final identification of the various pigments.

Additional comments and corrections are included in the attached file.

Author Response

Thank you for your suggestions and help to improve our manuscript.

The English language has been revised. All modifications in the manuscript have been reported in red.

A brief additional description of the pigments has been added in the conclusion.

Best Regards,

Maria Luisa Saladino

Reviewer 3 Report

Generally, I found the article interesting and well structured.

Below some issuues in question are listed:

17. The term "sample areas" (also in lines 67, 253, 267) is not clear. In Heritage science "sample" usually means either a portion of original material, collected in an invasive manner, or an  artificial construct (model sample = mock-up). So some explanation is needed. Were these areas chosen because they were the most representative or the most easily accessible?

115-122. Comparison of the area of the bishops hat and the red dress of the woman. Line 121-122: "XRF revealed the presence of mercury confirming the typical FCIR spectral response of this pigment used in mixture". You do not state clearly which pigment you are writing about (cinnabar?).

173. "The presence of sulphur (S) and strontium (Sr) in high quantities suggests the presence of the celestite (SrSO4), a mineral usually found associated with gypsum." I believe a citation is needed here. Moreover, I would appreciate a little more of a discussion about a popularity of celestite in Sicily (as far as I know it has been found in volcanic deposits from Mount Etna, but also can be commonly present in gypsum deriving from seawater precipitation). Anyway, XRF technique alone is not able to identify minerals and it should be stated that further research (eg XRD) can clarify the matter.

212. "Yellows: The signals of iron (Fe) and chromium (Cr) are present in both XRF spectra of the

P10 and P15 indicating the use of ochres." Ochres do not contain chromium and finding chromium in yellow indicates the use of 19th-20th c. restoration pigments (such as chrome yellow or strontium yellow). Since you haven't found Pb in spot P10, strontium yellow (Strontium chromate(VI), SrCrO4) could be probable. Anyway, associationg chromium with ochres is not correct and these results should be discussed more thoroughly. Moreover, in Fig. 12 I see titanium marked on both spectra (for P10 and P 15), but titanium is not listed in Table I. If present, titanium also indicates possible restoration area. In Fig 7a,b (IR imaging) the spot P15 looks like a dark patch. These data (XRF+IR+UV) should be interpreted in a more synergic manner in such questionable cases.

Figure 11 and 12., table I. The presence of chromium is discussed for blue and green areas  

but also red areas (33,35,40 and Red hat spot) need discussing. Maybe it would be worth to add a separate paragraph summarizing elements/pigments associated with restorations.

In table I column 2 (color) needs revising, since many spots are now not described with colour (e.g spots 22-28 and last 3 spots). Last 3 spots should also have added numbers and be marked in Fig. 9.

The text needs also a laguage revision and general polishing. Some English errors I found:

15. Should be "Its" instead of "His"

18. Should be "hand-held" instead of "handling"

51. Should be "detach it from the wall"

102. Should be "to map" instead of "to mapping"

148. Should be "artist's pallette" instead of "artist‘palette"

158. Should be "Localisation" instead of "Mapping"

161,222. Should be "carnation" or "complexion" instead of "incarnate"

225. Should be "copper" instead of "cupper"

225. "On the other hand" suggests that further part contradicts the identification of Cu (which is not the case).

248. Should be "mercury" instead of "Mercury"

259. Should be "to discover areas of restorations" instead of "to discover some repainting area"

260. Should be "details"

262. Should be "rich of details, despite of the fact that for a long time it was placed outside" instead of "even if"

Author Response

We are grateful to the reviewer for its suggestions and help to improve our manuscript. Please see below a point-by-point answer.

Generally, I found the article interesting and well structured. Below some issuues in question are listed:

17. The term "sample areas" (also in lines 67, 253, 267) is not clear. In Heritage science "sample" usually means either a portion of original material, collected in an invasive manner, or an  artificial construct (model sample = mock-up). So some explanation is needed. Were these areas chosen because they were the most representative or the most easily accessible?

The term “sample area” has been changed in “area”.

115-122. Comparison of the area of the bishops hat and the red dress of the woman. Line 121-122: "XRF revealed the presence of mercury confirming the typical FCIR spectral response of this pigment used in mixture". You do not state clearly which pigment you are writing about (cinnabar?).

The pigment is cinnabar. The word cinnabar has been substituted to the word pigment.

173. "The presence of sulphur (S) and strontium (Sr) in high quantities suggests the presence of the celestite (SrSO4), a mineral usually found associated with gypsum." I believe a citation is needed here. Moreover, I would appreciate a little more of a discussion about a popularity of celestite in Sicily (as far as I know it has been found in volcanic deposits from Mount Etna, but also can be commonly present in gypsum deriving from seawater precipitation). Anyway, XRF technique alone is not able to identify minerals and it should be stated that further research (eg XRD) can clarify the matter.

The references have been added. The following comment has been added in the text.

The presence of sulphur (S) and strontium (Sr) in high quantities suggests the presence of the gypsum and of celestite (SrSO4), a mineral usually found associated with gypsum. More precise information could be obtained by using a portable X-ray diffractometer for the investigation of minerals. The presence of gypsum can be an alteration product of calcium carbonate or a component used by the artist by propose [11-12]. The presence of strontium has been connected to the above second reason. On the other hand, the celestite in Sicily is often present in volcanic deposits from Mount Etna, but also can be commonly present in gypsum deriving from seawater.

212. "Yellows: The signals of iron (Fe) and chromium (Cr) are present in both XRF spectra of the P10 and P15 indicating the use of ochres." Ochres do not contain chromium and finding chromium in yellow indicates the use of 19th-20th c. restoration pigments (such as chrome yellow or strontium yellow). Since you haven't found Pb in spot P10, strontium yellow (Strontium chromate(VI), SrCrO4) could be probable. Anyway, associationg chromium with ochres is not correct and these results should be discussed more thoroughly. Moreover, in Fig. 12 I see titanium marked on both spectra (for P10 and P 15), but titanium is not listed in Table I. If present, titanium also indicates possible restoration area. In Fig 7a,b (IR imaging) the spot P15 looks like a dark patch. These data (XRF+IR+UV) should be interpreted in a more synergic manner in such questionable cases.

We agree with the reviewer comments: The text has been modified as:

Yellows: The signals of iron (Fe) is present in both XRF spectra of the P10 and P15 indicating the use of yellow ochres. The low counts of the chromium signal could be attributable to the presence of the modern pigment Chrome Yellow (PbCrO4) used for little retouching during the previous restorations also in this analysed area. The signals of lead (Pb) and of tin (Sn) are present in the P4 spectrum of the sleeve indicating the presence of the lead -tin yellow (Pb2SnO4 or PbSnO3). This yellow pictorial area is characterized by a totally different IR spectral behaviour (light grey tone) with respect the previous analysed area (P10 and P15 measurement points) which returned a greater absorption in IR (dark grey tones, Figures 6 and 7). The small peak of gold (Au) presents both in P4 and P10 spectra, can be related to some decoration in the drapery, verifying the preciousness of the fresco and provide important evidence of the original feature.

Figure 11 and 12., table I. The presence of chromium is discussed for blue and green areas but also red areas (33,35,40 and Red hat spot) need discussing. Maybe it would be worth to add a separate paragraph summarizing elements/pigments associated with restorations.

Some considerations about the pigments associated with the restorations have been added in the discussion and in the conclusions.

In table I column 2 (color) needs revising, since many spots are now not described with colour (e.g spots 22-28 and last 3 spots). Last 3 spots should also have added numbers and be marked in Fig. 9.

The table I and the Fig.9 have been modified.

The text needs also a laguage revision and general polishing. Some English errors I found:

15. Should be "Its" instead of "His"

18. Should be "hand-held" instead of "handling"

51. Should be "detach it from the wall"

102. Should be "to map" instead of "to mapping"

148. Should be "artist's pallette" instead of "artist‘palette"

158. Should be "Localisation" instead of "Mapping"

161,222. Should be "carnation" or "complexion" instead of "incarnate"               

225. Should be "copper" instead of "cupper"

225. "On the other hand" suggests that further part contradicts the identification of Cu (which is not the case).

248. Should be "mercury" instead of "Mercury"

259. Should be "to discover areas of restorations" instead of "to discover some repainting area"

260. Should be "details"

262. Should be "rich of details, despite of the fact that for a long time it was placed outside" instead of "even if"

English language has been revised.

Best Regards,

Maria Luisa Saladino

Round 2

Reviewer 1 Report

I put comments and suggestions in the new version of the manuscript: they should be visible as comments within the PDF (mainly as comments of highlighted text).

Let me know in case these are not visible.

My main criticism is still that I found poor attempt in combining data from different analytical measurements and in clearly discussing the interest for this multi-analytical approach. In my opinion, a discussion section is mandatory to improve the quality of the paper.

Author Response

Thank you for your suggestions and help to improve our manuscript.

All modifications in the manuscript have been reported in red. we put the answer to the reviewer's request on the attached pdf file.

Best Regards,

Maria Luisa Saladino
